# Negative-Pressure Wound Therapy (NPWT) in Horses: A Scoping Review

**DOI:** 10.3390/vetsci10080507

**Published:** 2023-08-06

**Authors:** Federica Cantatore, Eleonora Pagliara, Marco Marcatili, Andrea Bertuglia

**Affiliations:** 1Pool House Equine Clinic, IVC Evidensia, Crown Inn Farm, Fradley, Lichfield WS13 8RD, UK; marco.marcatili86@gmail.com; 2Department of Veterinary Sciences, University of Turin, 10095 Grugliasco, Italy; eleonora.pagliara@unito.it (E.P.); andrea.bertuglia@unito.it (A.B.)

**Keywords:** horses, NPWT, VAC therapy, wound, skin grafts

## Abstract

**Simple Summary:**

Wounds have a major impact on horses’ welfare and they can be a clinical challenge for practitioners. It is recognized that horses may experience delayed healing and the formation of exuberant granulation tissue. Negative-pressure wound therapy (NPWT) is a technique often employed in humans to enhance wound healing. It refers to the use of sub-atmospheric pressure obtained by a portable pump attached to a canister. However, the existing evidence for the effectiveness of NPWT remains uncertain in equine medicine. The aim of this review is to investigate NPWT applications and benefits concerning horses. The information obtained helps to provide recommendations for the use of this technique in practice. A review is performed and 24 manuscripts are considered. Fifteen manuscripts met the inclusion criteria. The focus of the articles was wound management, traumatic wounds, and surgical-site infections, including synovial structures as indications for NPWT. NPWT presents several advantages and few complications making it an attractive alternative to conventional wound management. However, randomized controlled trials should be performed to quantify the benefits and establish precise protocols in horses, which are lacking in the literature at present.

**Abstract:**

Obtaining a healthy wound environment that is conductive to healing in horses can be challenging. Negative-pressure wound therapy (NPWT) has been employed in humans to enhance wound healing for decades. The existing evidence for the effectiveness of NPWT remains uncertain in equine medicine. The aim of this review is to investigate NPWT applications and benefits in horses. A scoping review was performed according to the Preferred Reporting Items for Systematic Review and Meta-Analysis (PRISMA) guidelines for scoping reviews on three databases (PubMed, Web of Science-Thompson Reuters, and Wiley Online Library). Twenty-four manuscripts were considered. After removing duplicates, 17 papers underwent abstract screening. Of these, 16 + 1 (cited by others) were evaluated for eligibility according to PICOs, including no case reports/retrospective studies, four original articles, and three reviews. Fifteen manuscripts met the inclusion criteria. The focus of the articles was wound management; they included three reports of wounds communicating with synovial structures. Traumatic wounds and surgical-site infections are indications for NPWT. NPWT presents several advantages and few complications making it an attractive alternative to conventional wound management. However, randomized controlled trials should be performed to quantify the benefits and establish precise protocols in horses.

## 1. Introduction

Wound management in horses represents a significant challenge for equine practitioners. This is due to the intrinsic features of equine second-intention healing, which frequently leads to complications. For instance, distal limbs wounds are characterized by a weak acute inflammatory response leading to chronic inflammation and the development of exuberant granulation tissue [1,2]. This may result in delayed epithelialization, ineffective wound contraction, and prolonged or failed wound closure [3]. There are different wound-related factors that can negatively impact on healing in horses, such as types of wounds, age and location of the wound, involvement of structures other than skin, nature of the wound, previous treatments, and bioburden [4]. For these reasons, different modalities aiming at enhancing/improving wound healing and facilitating wounds management in equine practice are explored.

Negative-pressure wound therapy (NPWT), also commonly referred to as vacuum-assisted closure (VAC), has been used in human medicine for several years. It refers to the use of continuous or intermittent sub-atmospheric pressure (approximately 125 mmHg) [5]. There are multiple reported mechanisms of action of NPWT, including the reduction in dead space and the removal of exudates [6]. In turn, this reduces edema due to the extrusion of proinflammatory mediators, increase in dermal perfusion, and removal of wounds debris [7]. Secondarily, the mechanical stress of negative pressure stimulates the release of growth factors, increasing the cell proliferation and microdeformation of the wound surface enhancing the healing process via the formation of granulation tissue [8]. Furthermore, NWPT significantly reduces the number of bacteria within the wound bed [5]. The use of NPWT in equine medicine is limited in comparison with humans. There are several indications for using NPWT in horses. The introduction on the market of smaller and lighter devices has increased their use in clinical cases. NPWT has been reported to improve outcomes and reduce the number of complications [9]. NPWT is becoming a more established treatment to assist with the healing of challenging wounds in horses.

A scoping review is undertaken to evaluate the application and summarizes the evidence of the benefits of NPWT in equine medicine. The information obtained helps to provide recommendations for the use of this technique in practice.

## 2. Materials and Methods

A scoping review was conducted according to the Preferred Reporting Items for Systematic Reviews and Meta-Analysis (PRISMA) guidelines for scoping reviews [10]. A protocol was established and was not recorded on any base. The PRISMA checklist containing information relevant to this scoping review is reported in Appendix A. In March 2023, a systematic literature search was performed of potential studies investigating the use of negative-pressure wound therapy (NPWT, VAC^®^ therapy) in equine medicine published between March 1993 and 2023 by one of the authors (F.C.). The research question was: “which are the applications and clinical benefits of negative pressure wound therapy in horses?”.

Electronic databases were searched using the National Library of Medicine (PubMed), Web of Science-Thompson Reuters, and Wiley Online Library. A single search was used for each database using the advanced search function; the last search was performed on 29 May 2023.

For the NCBI PubMed database, the search string was the following:

(((“horse s” [All Fields] OR “horses” [MeSH Terms] OR “horses” [All Fields] OR “horse” [All Fields] OR (“equines” [All Fields] OR “horses” [MeSH Terms] OR “horses” [All Fields] OR “equine” [All Fields]) OR (“horse s” [All Fields] OR “horses” [MeSH Terms] OR “horses” [All Fields] OR “horse” [All Fields])) AND (“negative pressure wound therapy” [MeSH Terms] OR (“negative pressure” [All Fields] AND “wound” [All Fields] AND “therapy” [All Fields]) OR “negative pressure wound therapy” [All Fields] OR (“negative” [All Fields] AND “pressure” [All Fields] AND “wound” [All Fields] AND “therapy” [All Fields]) OR “negative pressure wound therapy” [All Fields])) OR (“negative pressure wound therapy” [MeSH Terms] OR (“negative pressure” [All Fields] AND “wound” [All Fields] AND “therapy” [All Fields]) OR “negative pressure wound therapy” [All Fields] OR (“vacuum” [All Fields] AND “assisted” [All Fields] AND “closure” [All Fields]) OR “vacuum assisted closure” [All Fields]) OR (“VAC” [All Fields] AND (“therapeutics” [MeSH Terms] OR “therapeutics” [All Fields] OR “therapies” [All Fields] OR “therapy” [MeSH Subheading] OR “therapy” [All Fields] OR “therapy s” [All Fields] OR “therapys” [All Fields]))) AND ((animal [Filter]) AND (31 March 1993:31 March 2023 [pdat]) AND (English [Filter])), 957, 17:51:43.

For the Web of Science database, the search string was the following:

(((ALL = (horse)) AND ALL = (negative pressure wound therapy)) OR ALL = (VAC therapy)) between 31 March 1993 and 31 March 2023.

For the Wiley Online Library, the search string was “horse” anywhere and “equine” anywhere and “horses” anywhere and “negative pressure wound therapy” anywhere and “vacuum-assisted closure” anywhere and “VAC therapy” anywhere between 1993 and 2023. The search was limited to journals in *Veterinary Medicine*.

References were imported from the search websites as titles and abstracts into an electronic spreadsheet (Microsoft Excel 16.54). The duplicates were removed; then, titles and abstracts were screened for relevance and eligibility for inclusion. Eligibility was determined according to the following criteria:The report was a peer-reviewed article.The report contained information about NPWT/VAC therapy use in horses.The publication was in English.If the articles were published between March 1993 and 2023.The articles were excluded if they were published in non-peer-reviewed journals, editorials, or congress proceedings where a full article was not available. Articles in a language other than English were excluded.

The articles were screened in two stages:

The first stage of screening was performed independently by two authors (F.C. and E.P.) based on the title and abstract. A consensus was reached through a vote between the four authors. Full-text papers were accessed from open access publications, library journal subscriptions, Turin University library, Nottingham University library, and open access sources. Papers that could not be retrieved and could not undergo second-stage screening were removed.

The second stage of the screening process was performed for the entire article. Eligibility was assessed following the objectives modified from “PICOs”: Population: horses; Intervention: wound treated with NPWT; and Outcome: follow-up assessment and complications. Additionally, reviews about equine wound care and management were included. The full-text studies were independently analyzed by two authors (F.C. and E.P.) and the relevant data were chartered, including type of study, species, sample size anatomical application of the NPWT, type of device used, disease that was treated, duration of application of NPWT, complications, and outcomes. Similarly, for the first selection, a consensus was obtained between all the authors.

## 3. Results

### 3.1. Selection of Sources of Evidence

The total number of papers that were retrieved was 4689, which was divided as follows:-NCBI-PubMed: 970.-Web of Science-Thompson Reuters: 2602.-Wiley Online Library: 1117.

After the irrelevant studies (human medicine studies, studies on other non-equid species, or not meeting the other inclusion criteria) (n = 4665), the removal of duplicates (n = 7), 17 abstracts underwent the first stage of screening. After one article was removed because it related to goats, a total of 16 papers underwent the second stage of screening considering the full-paper version of the studies. During full-paper screening, just 14 papers were included for eligibility and a new one was also included after being found as a citation in other papers, for a total of 15 articles [3,9,11,12,13,14,15,16,17,18,19,20,21,22,23].

### 3.2. Synthesis of Results

The selected articles included 8/15 case reports and retrospective articles, 4/15 original articles, of which just 2 included in vivo parts of the experiment, and 3/15 review articles (Figure 1).

In three manuscripts, NPWT was used to treat wounds communicating with synovial structures. These included three open, infected olecranon bursitis [11]; a report of a chronic multidrug-resistant wound leading to the infection of the antebrachiocarpal joint [12]; and a case series with three wounds communicating with the hock joints, two wounds communicating with the metacarpophalangeal joint, two wounds communicating with the proximal interphalangeal joint, 1 wound communicating with the carpal joint, one wound communicating with the calcaneal bursa, and four wounds communicating with the digital flexor tendon sheath (DFTS) [9].

The reported negative pressure used was −125 mmHg in a continued mode in most of the studies. However, in one report, the continued mode therapy was changed to intermittent-mode therapy after the first 24 h of treatment [11], and in another study the pressure was limited by the pump settings to −80 mmHg [3].

In the larger case series, to date, where NPWT was used to treat different types of distal limb wounds, the device was left in place for a period ranging from one to seven days (mean 4.5 days) prior to resetting the system. The total duration of the treatment was variable, ranging from 2–36 days (mean 11.5) [9]. The total duration of treatment in a controlled experimental study was limited to 6 days in contaminated wounds and 9 days in non-contaminated wounds [13]. When used to treat chronic infected olecranon bursitis, the bandage was changed every 3–4 days and NWPT was maintained in placed for 11 to 22 days [11]. In the case report describing the management of septic osteoarthritis of the antebrachiocarpal joint with a synovial cutaneous fistula, the bandage was changed every 2 days and the treatment continued for 12 days [12]. Additionally, Gemeinhardt and Molnar (2005) [14] and Florczyk et al. (2017) [15] opted to change the bandage every 3 to 4 days and VAC therapy was discontinued after 29 and 14 days, when the granulation was satisfactory (Table 1).

When NPWT was used in conjunction with skin grafts, foam was applied with the foal still under general anesthetic and the bandage was changed after 8, 11, and 14 days until removal at 19 and 11 days, and was removed at 18 days in the second case. The first case was a 9-month-old filly with a large traumatic non-healing skin laceration where two skin flaps already failed, and therefore a modified Meek transplantation technique was used associated with NPWT; the second was an adult Warmblood with cutaneous nodules on the saddle region, which were surgically excised and NPWT was applied [16]. In a similar case report described by Jordana et al. in 2011 [17], skin grafts were used together with NPWT for the treatment of a large non-healing wound on the dorsomedial and dorsolateral aspects of the metatarsus; the bandage was initially changed after 5 days, and the treatment discontinued.

Generally, NPWT was well-tolerated by the horses and very few complications were reported. Two horses, where NPWT was applied to treat a wound communicating with synovial structures (metacarpophalangeal joint and DFTS), were euthanized on humane grounds due to persistent and uncontrollable pain [9]. In one case, the NPWT system failed because the horse jumped forward shortly after its application. In 2 cases, the exudate clogged the tube; however, changing the setting to an alternate mode resolved the issue [9]. A few technical issues were encountered that resulted in a loss of sealing and therefore vacuum; however, this issue was encountered when a canister-free device was used. This device, in fact, was deemed to be effective just in the treatment of small animals but not in horses [3]. In other studies, cross-tying was used in cases of distal limb wounds to avoid the excessive movement of the horses impairing the sealing [9]. Topical reactions, such as cellulitis, edema, and mild erythema, were reported in just two studies [3,18]. Despite the general trend in NPWT providing beneficial effects on wound healing, the same results were not reported when NWPT was compared with standard wound care using calcium alginate dressings on contaminated and non-contaminated wounds that were surgically induced. This contrasts with what was reported by Askey et al. (2023) [18], where all the wounds healed with pleasing cosmetic and functional results, despite surgical-site infections with zoonotic, drug-resistant pathogens on the upper body of the horses. In equine surgery, NPWT is primarily used to enhance second-intention wound healing, but also to improve the acceptance of skin grafts taking over 75% or nearly up to 100% [16,17]. NPWT did not improve primary wound healing or prevent surgical-site infection when applied immediately after exploratory laparotomies in horses operated on because of the acute abdomen [20].

## 4. Discussion

Wounds have a major impact on the equine industry and horses’ welfare [24]. It is generally recognized that wounds on the distal limbs of horses have a slower rate of contraction and epithelialization, which often result in delayed healing and the formation of exuberant granulation tissue [25,26]. In turn, this can lead to aesthetical blemishes and, in the worst-case scenario, a loss of athletic function. When wounds communicate with synovial cavities, there can be a fatal outcome for the horse. At present, there is no single established method that can accelerate wound healing and prevent the formation of exuberant granulation tissue in equine medicine. Therefore, a comparison with modalities that have been proven to enhance wound healing in human patients may enable enhanced treatment options for horses [3]. Negative-pressure wound therapy is, at present, considered a first-line treatment in different types of wounds in humans [27], with the proven effect to reduce contaminations and surgical-site infections (SSIs) [28].

This review aimed at investigating the clinical use and benefits of NPWT in horses over the last 30 years to identify any potentially beneficial outcomes to indicate the effectiveness of the treatment. An attempt was made to identify the range of conditions for which NPWT might be usefully applied in horses. The use of three main databases allowed a comprehensive search of the potentially relevant literature, which, at this stage, remains limited. Fifteen manuscripts met the inclusion criteria of this scoping review. Many more articles have been published on the clinical use and efficacy of this technique in small animals [11]. However, the equine literature on the topic is limited. The first report was published nearly 20 years ago [14]. The limitations of this review were due to the restricted volume of the published literature on the topic and that it mainly consisted of case reports and case series. There is a substantial lack of randomized controlled trials in the equine literature where patients are randomly allocated to different treatment groups (i.e., comparing NPWT with different types of wound dressings, surgical wound debridement, or comparing one type of NPWT with another). The prospective research on NPWT in horses is lacking and is mainly performed on ex vivo specimens or surgically created wounds. These are far from ideal models for contaminated wounds encountered in equine practice. The primary outcomes of the studies were difficult to compare and inconsistent between studies as most of the manuscripts included were descriptive case series.

According to a large human review study, the therapeutic negative-pressure range is between −40 and −150 mmHg [29]. To date, there is conflicting evidence on the recommended settings in horses; however, the accepted standard negative pressure is −125 mmHg [22]. This is in line with the reported pressured used in the articles included in this review, where the pump was mainly used in a continuous mode. This setting is in fact the most efficient in preventing drain occlusions [30]. However, the intermittent setting seemed to improve granulation tissue formation [31]. In the original article using a lower negative pressure (i.e., 80 mmHg), several complications were encountered, including the lack of sealing and wrinkling of the bandage. The dressing used was considered as the cause of the negative result; however, in our opinion, utilizing a different pump may have improved the sealing [3].

It has recently been reported that NPWT is effective in reducing bacterial load contaminated wounds, especially when used in association with polyvinyl alcohol foam (NPWT-PVA) [20]. This characteristic is important in the management of cases of polymicrobial multidrug-resistant surgical-site infections and wounds without the concurrent administration of antimicrobials improving responsible antimicrobial stewardship. The advantage of containing septic exudate within a canister is that it limits the exposure to personnel and improves environmental biosecurity, which are important features to consider in a hospital [18]. NPWT may also seal surgical wounds, assisting primary closure and preventing nosocomial bacterial contamination [9].

The duration of NPWT in human medicine is approximately 4 to 7 days [32,33]. The ideal duration of treatment for NPWT in horses is not known and likely to be related to wound progression and the aim of the treatment [23]. Despite it being advisable to change the bandage every 2 to 3 days to reduce the growth of granulation tissue into the foam resulting in bleeding when it is removed, there are no studies on this subject for comparison [14]. The protocols used in the manuscripts considered were variable and longer intervals were often used, making the comparison difficult. Longer intervals may be beneficial to reduce costs, human handling of infected wounds, or disturbance of the grafts as suggested by Rijkenhuizen et al., (2005) [16].

Unrelenting pain was reported in two cases with wounds communicating with a synovial structure [9]. Although, in human medicine, NPWT is reported to be occasionally painful [34], this complication has never been reported in horses before. However, the severe pain experienced is more likely to be the result of the injury/septic synovial structure and not directly related to the NPWT application. NPWT is generally well-tolerated and can be applied to different types of wounds at different stages of healing.

Several NPWT units are available on the market at relatively affordable prices, which makes them easily accessible (Table 2). The limitation in its use is due to difficulties in securing the pump and canister to the horse. Therefore, Kamus et al. (2019) [3] investigated the effectiveness of a light-weight canister-free device (PICO^®^ from Smith & Nephew-Lachine, Canada), aiming at improving NPWT use in horses. Despite the system being well-tolerated, several difficulties were encountered with the dressing component making the device unsuitable for equine patients. Nevertheless, hooking the canister to the fluid line or a surcingle seemed to work well in horses, avoiding the necessity of cross-tying horses for a long period of time (Figure 2).

Haspeslagh et al. (2021) [13] provided disappointing results on the second-intention healing of surgically induced contaminated and non-contaminated wounds on distal limbs. However, it must be highlighted that the study was performed on a limited number of limbs and the model used in the study did not fully represent naturally occurring distal limb wounds (Figure 3 and Figure 4). Nevertheless, the treatment duration was shorter for contaminated wounds compared to non-contaminated (6 vs. 9 days), which seemed curious considering that the contaminated wounds tended to heal slower and with more discharge. Therefore, care needs to be exercised before translating the results into clinical case management until further investigations are performed. NPWT provided successful results in chronically infected olecranon bursitis (Figure 5), which did not respond to conservative treatment, with all the horses returning to their intended use [10], in wounds with exposed bones and communicating with synovial structures [9]. Wounds with extensive necrotic tissue and drainage seemed to be the ideal candidate for NPWT, even if specific reports are still lacking in the literature (Figure 6). Nevertheless, skin graft acceptance can be improved from the 60% rate historically reported in the literature [35] to nearly 100% after only 5 days of NPWT [17].

The fact that NPWT did not reduce the morbidity of wound complications after the exploratory laparotomy may be related to the fact that NPWT was applied after recovery from general anesthesia when the surgical incision was clean but not sterile. Additionally, the pattern chosen for the primary wound closure was a continuous simple suture pattern. The pattern may have inhibited the exudate removal, which is one of the features of NPWT [19]. Different results may be obtained if a group of horses with an established surgical-site infection and wound dehiscence following the exploratory laparotomy are included in a prospective study. 

## 5. Conclusions

To conclude, the reported benefits of NPWT in horses were:Limitation of the number of required bandage changes in cases of highly exudative wounds reducing costs and personnel involved [18].Increased hospital biosecurity by containing septic exudate. This is extremely important when dealing with infected wounds and surgical-site infections, especially when polymicrobial, multidrug-resistant bacterial pathogens are present [18]. Nosocomial infection may be limited, too [9].The amount of evacuated exudate can be easily monitored [30].Creation of hypoxic environment in the wound bed, which prevented aerobic bacteria replication and survival [21]. NPWT abolished the requirement of systemic antimicrobials, even in cases of highly infected wounds, facilitating responsible antimicrobial stewardship [18].NPWT facilitated the control and removal of bacteria biofilm [23].NPWT increased wound blood flow and granulation tissue formation, reducing the healing time and interstitial edema [5,31].Modulation of inflammatory and proliferative responses to the healing process [35].Increased skin graft acceptance [16,17].Limitation in wound retraction and facilitation of wound contraction pulling the margins together [14].

The reported downsides were:Requirements of specific equipment, including the cost of the initial purchase of the device and canisters.Requirement to keep the horse hospitalized during the treatment.

NPWT has a variety of applications and is associated with very few serious complications. Further investigations, including prospective studies, should be performed to establish a precise protocol for use in horses.

## Figures and Tables

**Figure 1 vetsci-10-00507-f001:**
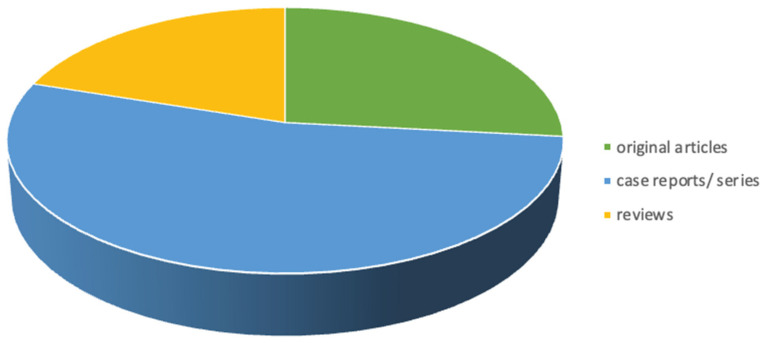
Pie chart showing type of publications included in scoping review (8/15 case series/retrospective studies, 4 original articles, 3 review articles).

**Figure 2 vetsci-10-00507-f002:**
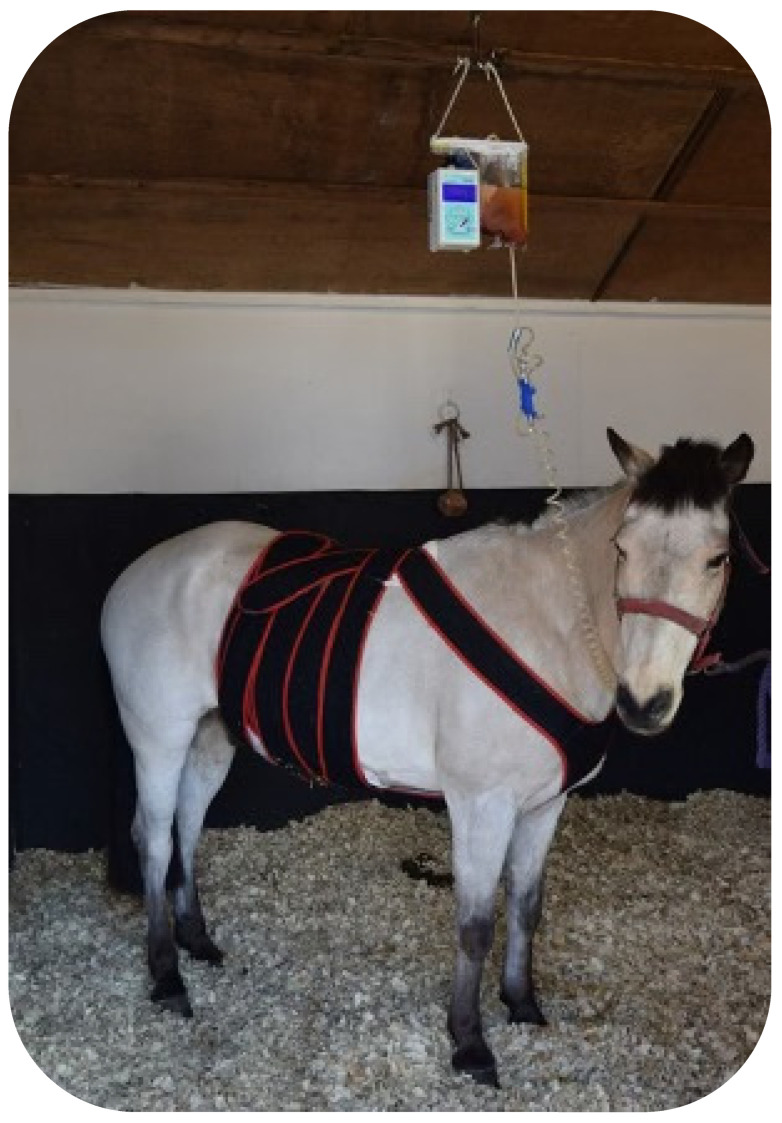
Picture of NPWT used for treatment of surgical-site infection post-exploratory laparotomy. Note the canister secured to a fluid line.

**Figure 3 vetsci-10-00507-f003:**
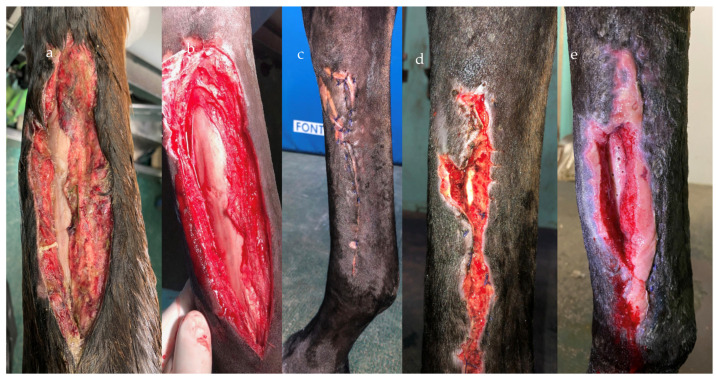
NPWT application on distal limb wounds with exposed bone. The contaminated wound on presentation (**a**). The same wound after surgical debridement (**b**). The wound was originally sutured and it was still closed 3 days after (**c**); however, the wound had a suture dehiscence (**d**) and the underling cannon bone wad exposed (**e**).

**Figure 4 vetsci-10-00507-f004:**
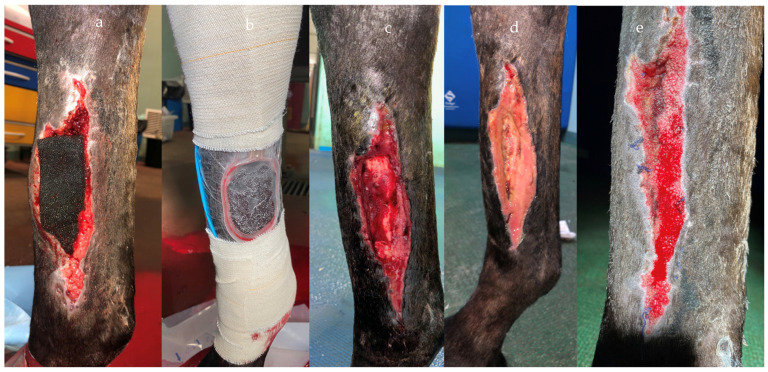
Same wound as Figure 3. NPWT was applied and maintained in place with a film and elastic bandage (**a**,**b**). Healthy granulation tissue developed from 9 days of treatment (**c**–**e**).

**Figure 5 vetsci-10-00507-f005:**
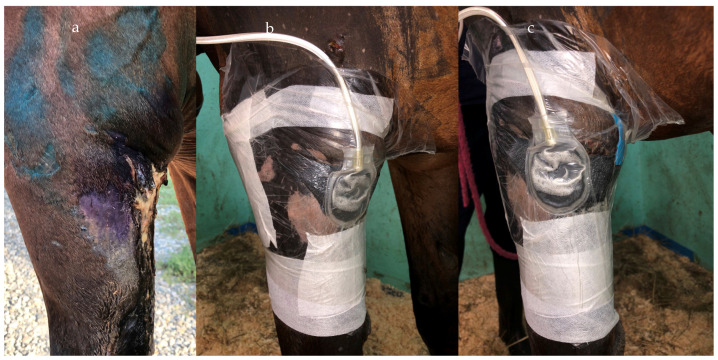
A traumatic wound on the palmar aspect of the elbow communicating with the olecranon bursa due to a fall (**a**) after the application of NPWT (**b**,**c**). The horse recovered successfully.

**Figure 6 vetsci-10-00507-f006:**
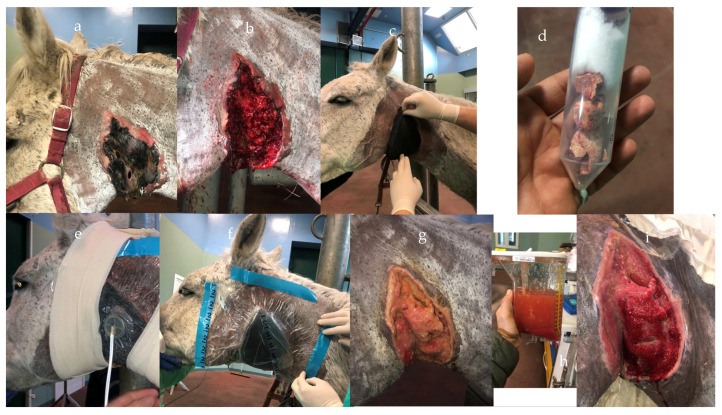
NPWT application in a horse with parotid gland duct rupture secondary to sialolithiasis. The horse presented extensive skin necrosis and exudate drainage (**a**). Wound after debridement (**b**), the parotid gland was exposed and NWPT was applied (**c**,**e**,**f**). Picture of the sialoliths obstructing the parotid gland, after removal using a transbuccal approach (**d**). An healthy granulation tissue started forming after 5 days (**g**,**i**). Picture of the fluid collected within the canister (**h**).

**Table 1 vetsci-10-00507-t001:** List of references with the duration of NPWT and frequency of the bandage changes during NPWT treatment.

Reference	Duration of the Treatment	Frequency of Bandage Change
Launois, T. et al., 2021 [9]	2–36 days (mean 11.5 days)	1–7 days (mean 4.5 days)
Haspeslagh, M. et al., 2021 [13]	6 days for contaminated wounds; 9 days in non-contaminated wounds	
Elce Y. et al., 2018 [11]	11 to 22 days	Every 3 to 4 days
Rettig, M. J., Lischer, C. J., 2017 [12]	12 days	Every 2 days
Gemeinhardt, K. D. and Molnar, J. A., 2005 [14]	29 days	Every 3 to 4 days
Florczyk, A., Rosser, J., 2017 [15]	14 days	Every 3 to 4 days
Rijkenhuizen et al., 2005 [16]	1st case: 19 days 2nd case: 18 days	1st case: bandage was changed at 8th, 11th, and 14th days after surgery 2nd case: bandage changed after 11 days
Jordana, M. et al., 2011 [17]	5 days	No bandage change
Gaus, M. et al., 2017 [19]	6 days	No bandage change unless the incision became uncovered or there was a loss of vacuum
Van Hecke, L. et al., 2016 [20]	24 h	6, 12, 18, 24 h after beginning ex vivo experiment
Askey, T. et al., 2022 [18]	4 to 70 days (mean 19.8 days)	Up to 7 days
Kamus, L. et al., 2019 [3]	Experimental wound study: 7 days	Experimental wound study: up to 3 to 4 times a day due to failure of maintaining the vacuum

**Table 2 vetsci-10-00507-t002:** List of the NPWT devices which were used in the studies considered in the review and their manufacturers.

Manufacturer/Supplier	NPWT System	Authors
KCI and Acelity, San Antonio, Texas, USA.	V.A.C. TracPadTM, V.A.C. Freedom	Elce, Y. A., et al., 2020 [11]
Allied Healthcare	Schuco portable suction unit	Askey, T., et al., 2023 [18]
KCI Medizinprodukte GmbH, Wiesbaden, Germany	Acti-V.A.C^®^	Gaus, M., et al., 2017 [19]
HAECO Medical Technology	VAC Pump	Florczyk A., Rosser, J., 2017 [15]
-KCI Medical B.V., Bergveste 12, 3992 DE Houten) -KCl, Kinetic Concepts Inc., USA	VAC^®^	-Rijkenhuizen, A. B. M. et al. [16], 2005 & Launois, T., et al., 2021 [9] -Jordana, M., et al., 2011 [17] & Gemeinhardt, K. D., Molnar, J. A., 2005 [14]
Smith & Nephew, Lachine Canada	PICO^®^	Kamus, L., et al., 2019 [3]
KCL Medical	VAC ATS	Haspelaslagh, M., et al., 2020 [13]
KCI Medizinprodukte GmbH, Wiesbaden, Germany.	VAC VeraFlo Instillation Therapy	Rettig, M. J., Lischer, C. J., 2017 [12]

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
