# Peer review of "Negative-Pressure Wound Therapy (NPWT) in Horses: A Scoping Review"

_vetsci, 2023, doi:10.3390/vetsci10080507_

Round 1

Reviewer 1 Report

The whole section on Materials & Methods were missing under Line 73

Author Response

The authors thank the reviewer for the feedback.

The entire paragraph MATERIAL & METHODS has been removed as  requested by Ms. Quenby Qu (Managing Editor) in the initial phase of the reviewing process. Please let me know if I can be of any further assistance.

Best Regards.

Reviewer 2 Report

Dear authors,

The article is well presented and explain. Interesting topic as, it is in fact somehow confusing weather this therapy actually helps or not.

Few comments:

The Materials and Methods part is empty?? Lines 75 to 90 seemed to be part of it, but certainly need to be corrected.

Line 162, lacks a coma ",": In turn, ...

The discussion is nicely presented and the conclusions as well.

I could not see the usefulness of the additional file to download as unpublished data?

There are some lines here and there that are not fully clear. As for example when mentioning this therapy applied to a foal under general anesthesia, as a reader, I am not aware of the characteristics of the clinical case, so it would be nice to shortly presented before, otherwise it is a bit confusing.

Line 162 missed a coma.

Author Response

Dear reviewer, 

please find your comments with appropriate answer in red.

The article is well presented and explain. Interesting topic as, it is in fact somehow confusing weather this therapy actually helps or not.

Few comments:

The Materials and Methods part is empty?? Lines 75 to 90 seemed to be part of it, but certainly need to be corrected.

The authors thanks the reviewer for the comment however the paragraph MATERIALS & METHODS was removed during  first stage of the review as suggested by Ms. Quenby Qu (Managing Editor).

Line 162, lacks a coma ",": In turn, ...

Thank you for noticing, the coma was added.

The discussion is nicely presented and the conclusions as well.

I could not see the usefulness of the additional file to download as unpublished data?

Thank you for your comment, however no supplement file is present in the manuscript.

There are some lines here and there that are not fully clear. As for example when mentioning this therapy applied to a foal under general anesthesia, as a reader, I am not aware of the characteristics of the clinical case, so it would be nice to shortly presented before, otherwise it is a bit confusing.

Thank you for your feedback, clinical cases characteristics have been added in lines 130-134 and 134-137.

Please let me know if your require further clarifications.

Round 2

Reviewer 1 Report

Based on the author's reply, I have no further comments.